**Data Availability Statement:** The data underlying the results presented in the study can be found in the Supporting Information.

**Funding:** The authors received no specific funding.

# Protein quantification and enzyme activity estimation of Pakistani wheat landraces

**Iram Mughal[1], Yusra Shah[1], Shamaila Tahir[1], Waseem Haider[1], Muhammad Fayyaz[2], Tayyaba Yasmin[1], Maryam Ilyas[1], Sumaira Farrakh[1]***

**1** Department of Biosciences, COMSATS University Islamabad, Islamabad, Pakistan, **2** National Agricultural Research Center, Islamabad, Pakistan

* sumaira.farrakh@comsats.edu.pk

## Abstract

Wheat is a major food grain in Pakistan having a prominent role in agriculture as well as the economic status of the country. In the current study, seeds of 99 wheat landraces were characterized for the quantification of seed storage proteins (Albumins, Globulin, Gliadins, and Glutenin), enzyme activities of antioxidant enzymes i.e. Ascorbate peroxidase (APX), Catalase (CAT), Superoxide dismutase (SOD), Peroxidase (POD), one hydrolytic enzyme Protease (PROT) and non-enzymatic antioxidant enzyme Ascorbic acid (AsA). The landraces were categorized into low, medium, and high based on protein concentration and enzymes activities/content. The majority of the landraces were placed in the medium category. However, for the AsA parameter majority of the landraces were placed in the low category. The highest concentration of total extracted protein (184.88±0.7 mg/g. wt.), globulins (21.35 ±0.43 mg/g. wt.) and glutenin (20±0.04 mg/g. wt.) as well as the high activity of SOD (303 ±16.80 Units/g. wt.), and Ascorbic acid (533±36.1 Units/g. wt.) was identified in the wheat landrace "11757" collected from district Panjgur (Balochistan). The wheat landrace "11760", collected from district Kech (Balochistan), contained the highest albumins concentration (65.42±0.02 mg/g. wt.) and highest activity for CAT (589.5±61.20 Units/g. wt.). The highest activity of POD (32341± 91.3) and PROT was observed in seeds of the wheat landrace "11618" collected from the Gilgit Baltistan region of Pakistan. The principal component analysis showed that the great variations existed for the tested parameters among the wheat landraces. The landraces with a high concentration of seed storage proteins and antioxidant enzyme activities can be used for breeding purposes to improve the nutrimental quality of wheat cultivars.

## Introduction

Wheat (*Tritium aestivum L.*) is the most dominant crop all over the world and it is among the crucial three cereal crops that supply 20% of the gross energy needed in human food [1]. Wheat cultivation is predominantly concentrated in China, the USSR, Turkey, Ukraine, Australia, USA, India, and Pakistan, which accounts for about 80% of the worldwide wheat cultivation [2]. Out of one hundred twenty-one wheat-producing countries, Pakistan is the 8th largest

**Competing interests:** The authors have declared that no competing interests exist.

wheat producer, sharing 3.17% of the world wheat production from only 3.72% of the wheat-growing area. In Pakistan, it is a major food grain contributing 72% of daily caloric intake with per capita wheat consumption of around 124 kg per year, one of the highest in the world [3].

The wheat flour is used in different foods and its utilization is mainly determined by its protein content. The proteins make up 8–20% of mature wheat grain. Following the sequential Osborne extraction procedure, albumins, globulins, gliadins, and glutenin can be isolated. Albumins and globulins of wheat endosperm represent 20% to 25% of total grain proteins [4]. Wheat is a cheap source of essential amino acids (which are not synthesized in body), good quality minerals, vitamins, and vital dietary fibers to the human diet [5]. Besides this, it is also considered as natural source of both enzymatic and non-enzymatic antioxidants. The enzymatic antioxidants include superoxide dismutase (SOD), glutathione reductase (GR), and ascorbate peroxidase (APX), catalase (CAT) and peroxidase (POD) while non-enzymatic antioxidants include vitamin C (tocopherols and tocotrienols), vitamin E, and carotenoids [6].

Oxidative damage to enzymes and DNA is prevented by these antioxidants which, directly react with Reactive Oxygen Species [7]. Reduced risk of oxidative-stress related chronic diseases and age-related disorders, such as cardiovascular diseases, carcinogenesis, type II diabetes, and obesity were found to be associated with the consumption of whole-grain wheat flour and products [8].

Recently a number of breeding programs have initiated the selection and development of wheat varieties with high yield and improved seed quality for specific end-product quality. As a consequence of this, not only in Pakistan but throughout the world modern wheat cultivars are often genetically similar, with a rather compromised genetic base [9]. The sole reason for this compromised genetic make is the reliance on a limited number of parent lines. A report has suggested that due to this reliance on selected varieties, the population size of wheat has been reduced by 6% [10] which limits the improvement of many traits in wheat. Therefore, there is a dire need to explore wheat germplasm for the identification of lines with unique and improved nutritional qualities.

Wheat landraces are traditional wheat varieties developed by combination of both humans and natural selection. These are not only well adapted to the local environmental conditions and management practices [11] but also offer a valuable source to broaden the genetic base of cultivated wheat for various traits including nutritional qualities. The use of wheat landraces for direct crossing and introgression of adaptive traits is an attractive breeding strategy.

Thousands of accessions of wheat landraces have been deposited in different Gene Banks. All of these accessions cannot be used in the breeding programs. Therefore, prior to their utilization in the breeding programs, these landraces must be screened for various traits, for the selection of most suitable accessions. One of these traits is seed quality, which is mainly based on seed storage proteins and the presence of phyto-chemicals and antioxidants. Keeping in view these parameters, the current study was designed to screen 99 wheat landraces collected from different agro-ecological zones of Pakistan for the selection of landraces with desirable traits.

The main aim of this research was to examine the differences in the protein content and antioxidant activity among the wheat landraces to identify the landraces with superior nutritional quality.

## Materials and methods

### Wheat germplasm

Seeds of 99 wheat landraces were acquired from the Plant Genetic Resource Institute (PGRI) (S1 Table). Flour of these seeds was prepared by grinding these seeds in pestle and mortar. Three replicates were used for each landrace.

## Total extracted proteins estimation

For total protein content, 100mg of seeds were grinded in a buffer composed of 50 mM potassium phosphate buffer (pH 7.0). The homogenate was centrifuged for 20min at 10,000rpm at 4 ˚C. The supernatant was transferred to a clean Eppendorf and labelled as TPC-1.

## Differential protein estimation

**Albumin (ALB).**   Seed storage proteins were extracted by following the Osborne fractionation method [12]. 100mg of grinded seeds were extracted with autoclaved distilled water (500 µl) for 30 minutes at 4 ˚C, vortexed for 1 minute, at 10 minutes interval. Centrifugation was done at 2,000 rpm for 5 minutes. The supernatant was poured into separate Eppendorf, named ALB-1. The pellet was vortexed with autoclaved distilled water (400 µl) for 1 minute, then allowed to be settled for 5 minutes, centrifuged as the previous step and supernatant was mixed with ALB-1. This step was repeated, and the supernatant was mixed with ALB-1. The same procedure was adopted for other replicates.

**Globulin (GLOB).**   The pellet from the previous step was dissolved with 400 µl of NaCl solution (0.5M) for 30 minutes at 4 ˚C, vortexed for 1 minute, after every 10 minutes. Centrifugation was done at 2,000 rpm for 5 minutes. The supernatant was poured into a new Eppendorf and saved as GLOB-1. 400 µl of 0.5M NaCl was added to the pellet and vortexed for 1 minute, then allowed to be settled for 5 minutes and centrifuged as the previous step. This step was repeated twice, and the supernatant was mixed with GLOB-1. The pellet was washed with 400ul of autoclaved distilled water to decrease the presence of the salt from the pellet. The same procedure was adopted for other replicates.

**Gliadin (GLI).**   The water-washed pellet from the previous step was dissolved with 400 µl of 70% ethanol for 30 minutes at 4˚C, vortexed for 1 minute, at 10 minutes interval. The dissolved pellet was centrifuged at 2,000 rpm for 5 minutes. The supernatant was poured into a new Eppendorf tube and named as GLI- 1. The pellet was then vortexed with 400 µl of 70% ethanol for 1 minute, then allowed to be settled for 5 minutes and centrifuged as the previous step. This step was repeated twice, and the supernatant was mixed with GLI-1. The same procedure was adopted for other replicates.

**Glutenin (GLU).**   The pellet from the previous step was dissolved with 400 µl of 50% 1-propanol + 1% Beta mercaptoethanol (BME) for 30 minutes at 4˚C, vortexed for 1 minute, after every 10 minutes. Centrifugation was done at 2,000 rpm for 5 minutes; then the supernatant was poured off into the separate Eppendorf and named GLU-1. This step was repeated twice, and the supernatant was mixed with GLU-1. The same procedure was adopted for other replicates.

**Quantification of proteins.**   Quantification of extracted proteins (total proteins, albumins, globulins, gliadins, and glutenin) was conducted using a spectrophotometer, and BSA (Bovine Serum Albumin) was used as a standard. Six different standard solutions were prepared. These standards contained 0, 2.5, 5, 10, 15, 20µl of BSA stock (1mg/ml) respectively, in 1ml of Bradford reagent. A total 2µl of each protein sample was mixed with 1ml of Bradford reagent. All of the tubes were inverted, and absorbance was measured using UV-VIS Spectrophotometer (Thermo Fisher Scientific) at 595nm [13]. The standard linear curve of six points was created by using MS excel and concentrations of protein samples were calculated.

## Enzymes activity estimation

**Ascorbate Peroxidase (APX).**   To determine the activity of the ascorbate peroxidase, the homogenization of wheat seeds was done in potassium phosphate buffer (50mM) [14]. The assay buffer contained potassium phosphate buffer (200mM), ascorbic acid (10mM), and

EDTA (0.5M). An activity solution was prepared to estimate ascorbate peroxidase from ascorbic acid (10 mM), EDTA (0.5 M), and $KH_2PO_4$ buffer (200mM), hydrogen peroxide (1 ml) and 50μl of supernatant. The absorbance of the reaction mixture was taken at 290 nm after every 30 seconds by using a UV-VIS spectrophotometer. A decrease in absorbance indicated the oxidation of ascorbic acid [15].

**Catalase (CAT).** To determine catalase activity, the homogenization of wheat grains was done in a mixture of potassium phosphate buffer (50 mM) and BME (1 mM). The activity solution contained phosphate buffer (50 mM), $H_2O_2$ (59 mM), and an enzyme extract (100 ul). The absorbance of the reaction mixture was taken at 240 nm after every 20 seconds by using a UV-VIS spectrophotometer. A decrease in absorption indicated the activity of the solution. 1U of catalase activity is the 0.01 change in absorbance of activity solution per minute [16].

**Superoxide Dismutase (SOD).** To determine superoxide dismutase (SOD) activity, the homogenization of seed samples was done in potassium phosphate buffer (50mM), EDTA (0.1 mM), and BME (1 mM) [15]. The SOD activity was quantified in terms of its capacity to hinder the decrease of NBT photochemically [17]. 1U of an enzyme activity equals to 50% NBT inhibition.

**Peroxidase (POD).** To determine the peroxidase (POD) activity, the homogenization of seed samples was done in potassium phosphate (50 mM), EDTA (0.1 M), and BME (1 mM) [18]. The assay solution was made by mixing $dH_2O$ (545μl), phosphate buffer (200 mM), guaiacol (200 mM), hydrogen peroxide (400 mM), and an enzyme extract (15μl). The addition of the enzyme extract started the reaction. Absorbance was taken at 470 nm at 20 seconds interval by using a UV-VIS spectrophotometer. 0.01change in absorbance per minute was designated as 1U of peroxidase activity. The enzyme activity was expressed in terms of the weight of the seed.

**Protease (PROT).** To determine the activity of the protease, the homogenization of the seeds was done in potassium phosphate buffer (50 mM). Protease activity was measured by the casein digestion assay [19]. Change of 0.001/ minute in absorbance at 280 nm was defined as 1U of protease. It is the amount of enzyme that liberates fragments which are acid soluble.

**Ascorbic Acid (AsA).** For the determination of ascorbic acid activity in the wheat grains, 2,6-dichloroindophenol was used in the reaction. In this reaction, molecules of DCIP were reduced to $DCIPH_2$ by the action of vitamin C, and that reduction was recorded as a drop in the absorbance at 520 nm. A standard curve was drawn by using a series of known ascorbic acid concentrations. The Ascorbic acid concentration in unknown samples was found out by a simple linear regression equation [20].

## Statistical analysis

All the data (differential protein and enzyme activity) was separately reported as mean ± SD. Principal Component Analysis (PCA) of the data was done by XLSTAT software (Version 2019).

## Results

### Total Extracted Proteins estimation (TEP)

The landraces were categorized into three groups; low, medium, and high on the basis of total protein concentration per gram of wheat seed (Table 1). Twenty-six percent (26) of the tested landraces were placed in the group showing high concentrations for total extracted proteins. The concentration of TEP ranged from 151.40 to 184 mg/g s. wt. The landrace "11757" showed the highest concentration of TEP (184.88±2.5 mg/g. wt.). Sixty percent (60) of the tested

**Table 1. Categorization of wheat landraces in low, medium, and high value for seed storage proteins concentration and antioxidant enzyme activities [32].**

| Acc# | Location | TEP | Alb | Glob | Gli | Glu | Ascorbate peroxidase | Catalase | Protease | Superoxide dismutase | Peroxidase | Ascorbic acid |
|---|---|---|---|---|---|---|---|---|---|---|---|---|
| 11526 | Sindh | M | M | L | M | M | M | L | M | M | M | L |
| 11528 | Balochistan | H | H | L | H | M | M | H | H | L | M | L |
| 11534 | Balochistan | L | M | L | L | L | H | M | L | L | H | L |
| 11535 | Balochistan | M | M | L | M | L | H | H | M | M | M | H |
| 11538 | Balochistan | M | M | L | M | L | L | M | L | M | H | M |
| 11539 | Balochistan | L | M | L | L | L | L | H | L | L | M | L |
| 11540 | Balochistan | M | M | L | L | M | H | M | H | L | M | H |
| 11543 | Balochistan | L | M | L | L | L | M | M | H | M | M | H |
| 11545 | Balochistan | M | M | L | M | L | M | M | M | M | M | H |
| 11546 | Balochistan | L | M | L | L | L | M | H | L | L | L | M |
| 11548 | Balochistan | M | L | M | M | L | M | M | L | L | M | L |
| 11549 | Balochistan | M | L | L | M | L | L | M | L | L | M | L |
| 11550 | Balochistan | M | M | H | M | H | M | H | L | M | M | L |
| 11551 | Balochistan | M | L | M | H | M | L | H | L | L | M | L |
| 11552 | Balochistan | H | L | 0 | M | H | M | M | M | M | M | L |
| 11553 | Balochistan | M | M | 0 | M | H | M | M | L | M | M | L |
| 11554 | Balochistan | M | M | M | H | L | L | H | L | L | M | L |
| 11555 | Balochistan | H | M | M | M | L | M | M | L | H | M | L |
| 11556 | Balochistan | H | M | M | H | H | M | M | L | H | M | L |
| 11557 | Balochistan | M | M | L | M | L | M | M | L | L | M | L |
| 11558 | Balochistan | H | H | H | H | H | L | H | L | L | M | L |
| 11560 | Balochistan | M | M | L | M | L | H | H | L | L | L | M |
| 11561 | Gilgit-Baltistan | L | M | L | L | L | L | M | M | M | H | L |
| 11562 | Gilgit-Baltistan | L | M | L | L | L | M | L | H | M | H | M |
| 11563 | Gilgit-Baltistan | M | M | L | M | L | M | H | L | H | M | H |
| 11564 | Gilgit-Baltistan | M | M | L | M | L | H | H | M | L | M | L |
| 11565 | Gilgit-Baltistan | M | M | L | M | M | L | H | M | L | M | L |
| 11566 | Gilgit-Baltistan | L | M | L | L | L | H | H | L | H | L | L |
| 11568 | Gilgit-Baltistan | M | M | L | M | L | H | H | H | M | M | H |
| 11569 | Gilgit-Baltistan | L | M | L | L | L | H | H | L | L | L | M |
| 11570 | Gilgit-Baltistan | M | M | L | M | L | M | M | M | L | M | L |
| 11571 | Gilgit-Baltistan | M | M | L | M | L | M | M | L | H | L | L |
| 11572 | Gilgit-Baltistan | L | M | L | L | L | M | H | M | L | M | L |
| 11573 | Gilgit-Baltistan | M | M | L | M | L | H | H | M | L | M | H |
| 11574 | Gilgit-Baltistan | M | M | L | M | L | H | H | H | L | M | L |
| 11576 | Gilgit-Baltistan | M | M | L | M | M | M | H | L | L | M | L |
| 11577 | Gilgit-Baltistan | M | M | L | M | L | M | M | L | L | M | L |
| 11578 | Gilgit-Baltistan | M | M | L | M | L | H | H | H | L | M | L |
| 11580 | Gilgit-Baltistan | L | M | L | L | L | H | H | H | L | M | L |
| 11581 | Gilgit-Baltistan | L | M | L | L | L | M | H | H | L | M | L |
| 11582 | Gilgit-Baltistan | M | M | L | M | L | H | H | H | L | M | L |
| 11583 | Gilgit-Baltistan | M | L | M | M | L | M | H | L | L | M | L |
| 11584 | Gilgit-Baltistan | H | L | M | H | L | M | M | L | L | M | L |
| 11586 | Gilgit-Baltistan | M | L | M | M | M | M | H | L | L | M | L |
| 11587 | Gilgit-Baltistan | M | L | L | M | M | L | M | M | L | M | L |
| 11591 | Gilgit-Baltistan | M | L | M | M | L | L | M | L | L | M | L |
| 11593 | Gilgit-Baltistan | M | M | M | M | M | M | M | L | L | M | L |

(*Continued*)

**Table 1.** (Continued)

| Acc# | Location | TEP | Alb | Glob | Gli | Glu | Ascorbate peroxidase | Catalase | Protease | Superoxide dismutase | Peroxidase | Ascorbic acid |
|------|----------|-----|-----|------|-----|-----|----------------------|----------|----------|----------------------|------------|---------------|
| 11594 | Gilgit-Baltistan | H | M | M | H | M | M | H | L | L | M | L |
| 11595 | Gilgit-Baltistan | H | M | M | H | M | M | H | L | L | M | L |
| 11596 | Gilgit-Baltistan | M | L | M | M | L | M | M | L | L | M | L |
| 11597 | Gilgit-Baltistan | M | L | M | M | M | M | M | L | L | M | L |
| 11598 | Gilgit-Baltistan | M | L | L | M | L | M | H | L | L | M | L |
| 11599 | Gilgit-Baltistan | M | L | L | M | L | M | M | L | L | M | L |
| 11600 | Gilgit-Baltistan | H | M | M | H | H | M | H | M | L | M | L |
| 11601 | Gilgit-Baltistan | H | M | L | M | M | M | M | L | L | M | L |
| 11602 | Gilgit-Baltistan | M | M | L | M | H | M | M | L | L | M | L |
| 11603 | Gilgit-Baltistan | H | M | L | H | M | L | M | L | L | M | L |
| 11604 | Gilgit-Baltistan | H | M | L | H | L | M | M | L | L | M | L |
| 11607 | Gilgit-Baltistan | M | M | L | M | L | L | H | L | L | M | L |
| 11611 | KPK | M | L | L | M | M | L | M | L | L | M | L |
| 11612 | KPK | H | M | H | M | H | M | M | H | L | M | L |
| 11613 | KPK | L | L | L | M | L | M | H | M | L | M | L |
| 11614 | KPK | M | L | L | M | L | M | H | M | L | M | L |
| 11615 | Gilgit-Baltistan | H | L | L | M | L | M | H | M | L | M | L |
| 11618 | Gilgit-Baltistan | H | M | H | H | L | M | M | H | L | M | L |
| 11622 | Syria | M | H | L | M | H | H | M | M | L | M | L |
| 11623 | Syria | M | L | L | M | L | M | H | H | L | M | L |
| 11624 | Punjab | M | L | L | M | L | H | H | M | L | M | L |
| 11625 | Punjab | M | L | L | M | L | M | M | M | L | M | L |
| 11626 | Punjab | H | M | H | M | M | M | H | H | L | M | L |
| 11649 | Punjab | H | M | L | H | H | M | H | M | L | M | L |
| 11650 | Punjab | M | L | L | M | M | M | M | L | L | M | L |
| 11651 | Punjab | M | L | L | M | L | M | H | L | L | M | L |
| 11652 | Punjab | M | L | L | M | M | M | M | L | L | M | L |
| 11653 | Punjab | M | L | L | M | M | L | H | L | M | M | L |
| 11654 | Punjab | M | L | L | M | H | M | M | L | M | M | L |
| 11655 | Punjab | H | M | L | M | H | M | M | L | M | L | L |
| 11656 | Punjab | M | L | L | M | L | M | M | L | M | M | L |
| 11657 | Punjab | M | L | L | M | H | M | M | L | H | L | L |
| 11658 | Punjab | M | L | L | M | M | M | H | M | L | M | L |
| 11681 | Punjab | M | L | L | M | M | M | M | L | L | L | L |
| 11682 | Punjab | L | L | L | M | L | M | M | L | L | M | L |
| 11683 | Punjab | H | M | H | M | H | L | M | L | L | M | L |
| 11684 | Punjab | M | L | L | M | M | M | M | L | M | M | L |
| 11685 | Punjab | M | L | L | M | M | M | M | L | H | M | L |
| 11686 | Punjab | M | L | L | M | L | M | M | M | M | M | L |
| 11687 | Punjab | H | L | H | M | H | L | H | L | L | M | L |
| 11688 | Punjab | M | L | L | M | H | L | M | L | M | M | L |
| 11689 | Punjab | M | L | L | M | H | M | M | L | M | L | L |
| 11690 | Punjab | H | M | L | H | M | M | M | L | L | L | L |
| 11754 | Balochistan | M | L | L | M | L | H | M | L | L | L | M |
| 11755 | Balochistan | M | L | H | M | L | M | H | M | M | H | L |
| 11757 | Balochistan | H | H | H | H | H | H | M | M | H | M | H |
| 11758 | Balochistan | H | M | H | H | H | H | H | M | M | H | M |

(*Continued*)

**Table 1.** (Continued)

| Acc# | Location | TEP | Alb | Glob | Gli | Glu | Ascorbate peroxidase | Catalase | Protease | Superoxide dismutase | Peroxidase | Ascorbic acid |
|---|---|---|---|---|---|---|---|---|---|---|---|---|
| 11760 | Balochistan | H | H | H | H | H | M | H | M | L | M | L |
| 11761 | Balochistan | M | L | L | M | L | M | M | L | L | M | H |
| 11762 | Balochistan | M | M | L | M | L | M | M | L | H | L | L |
| 11763 | Gilgit-Baltistan | M | M | L | M | L | H | H | H | M | M | H |
| 11767 | Gilgit-Baltistan | H | M | L | M | L | H | M | H | L | L | M |

Total Extracted Proteins (TEP): Low = <110 mg/g. wt., Medium = 111-150mg/g. wt., High = 151–185 mg/g. wt.

Albumin (Alb): Low = <40mg/g. wt., Medium = 41-50mg/g. wt., high = 51–55 mg/g. wt.

Globulin (Glob): Low = <13 mg/g. wt., Medium = 14–17 mg/g. wt., High = 18-21mg/g. wt.

Gliadins (Gli): Low = <55 mg/g. wt., Medium = 56–80 mg/g. wt., High = 81–93 mg/g. wt.

Glutenin (Glu): Low = <12 mg/wt., Medium = 13–16 mg/g. wt., High = 17–20 mg/g. wt.

Ascorbate peroxidase (APX) (Units/g s. wt.): Low = <400, Medium = 401–932, High = 1000–1700.

Catalase (Units/g s. wt.) (CAT): Low = <100, Medium, 101–367, High, 368–634.

Protease (Units/g s. wt.) (PROT): Low = <6000, Medium, 6000–8800, High, 8801–11183.

Superoxide dismutase (SOD) (Units/g s. wt.): Low = 110, Medium = 171–250, High = 251–310.

Peroxidase (Units/g s. wt.) (POD): <13000, Medium = 13001–25000, High = 25001–33000.

Ascorbic acid (μg /g s. wt. (AsA)): <640, Medium = 641–681, High = 682–713.

landraces were placed in a group showing medium concentrations for total extracted proteins. The concentration in the group ranged from 110.91 to 145.01 mg/g. wt. Thirteen percent (13) of the tested landraces showed the low concentration of total extracted proteins ≤110. Lowest total extracted proteins content was found in the landrace "11613" (98.05± 3.4 mg/g. wt.) and "11561" (98.05± 2.9 mg/g. wt.) (Fig 1, S1 Fig).

## Differential protein estimation

**Albumin.**   The landraces were categorized into three groups; low, medium, and high based on albumin concentration per gram of wheat seed (Table 1). Five percent (5) of the tested landraces showed a high concentration of albumin. The highest concentration of albumins (55.42±0.02 mg/g. wt.) was found in the landrace "11760". Thirty-one percent (31) of the tested landraces were grouped in the medium category with albumin concentration ranged between 41–50 mg/g. wt. Forty-eight percent (48) of the tested landraces were classified in the low category for albumin concentration. The lowest albumin concentration (30.12±0.20 mg/g. wt.) was found in the landrace "11561" (Fig 1, S2 Fig).

**Globulin.**   The landraces were categorized into three groups; low, medium, and high on the basis of globulin concentration per gram of wheat seed (Table 1). Seventeen percent (17) of the landraces were placed in a high category for globulin. The concentration ranged from 18–21 mg/g. wt. and the highest concentration of globulins (21.35±0.43 mg/g. wt.) was found in landrace "11757". Sixty-one percent (61) of the wheat landraces were grouped in the medium category with globulin concentration ranged between 14–17 mg/g. wt. Thirty-nine percent (39) of the wheat landraces were classified in the low category for globulins. The lowest globulins content was found in landrace "11546" (10.34±1.80 mg/g. wt.) (Fig 1, S3 Fig).

**Gliadin.**   The landraces were categorized into three groups; low, medium, and high on the basis of gliadin concentration per gram of wheat seed (Table 1). Seventeen percent (17) of the landraces were placed in the high category. The concentration ranged from 81–93 mg/g. wt. Landrace "11758" showed the highest concentration for gliadin (93.27±1.9 mg/g. wt.). Sixty-one percent (61) of landraces were grouped in the medium category with gliadin

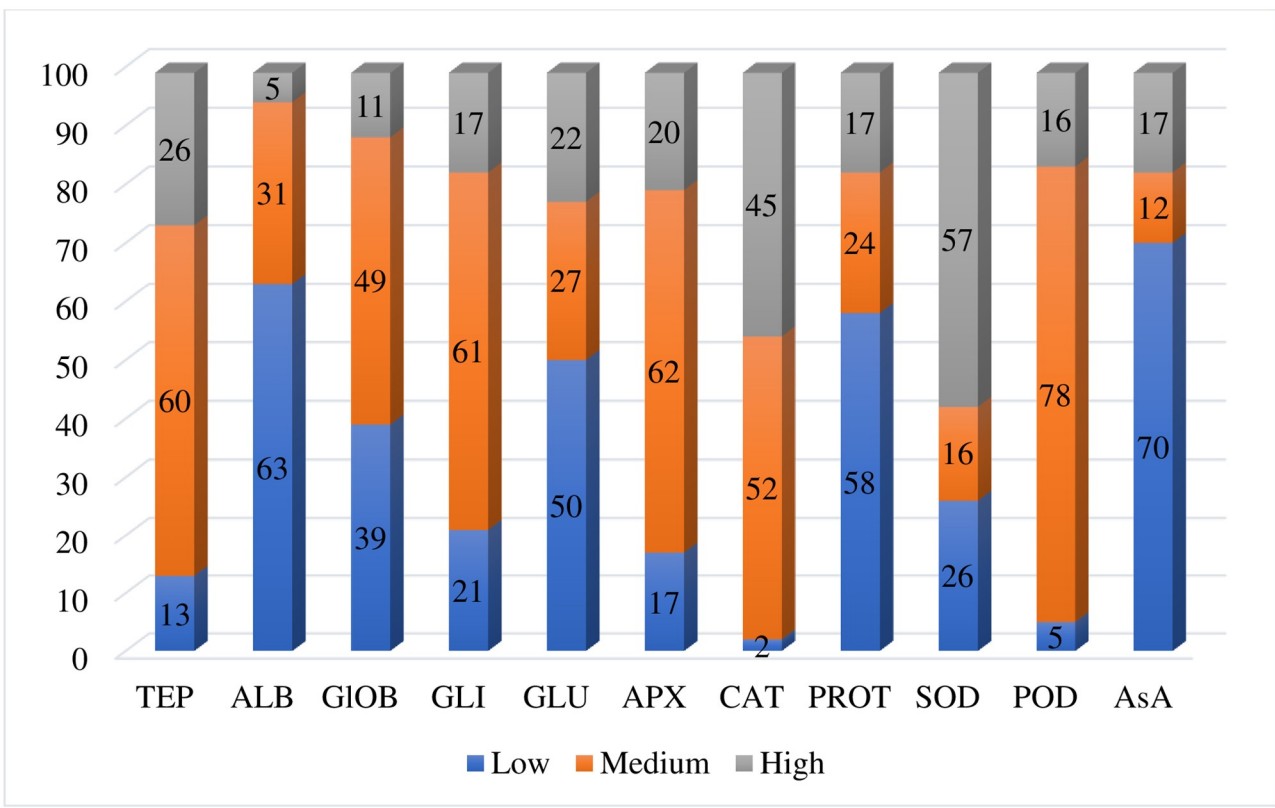

**Fig 1. Frequency distribution of wheat landraces in low, medium and high categories for selected parameters.**

concentration ranged between 56–80 mg/g. wt. Twenty-one percent (21) of the landraces were classified in the low category for gliadins. The lowest gliadin concentration was found in the landrace "11613" which was 45.6±2.5 mg/g. wt. (Fig 1, S4 Fig).

**Glutenin.** The landraces were categorized into three groups; low, medium, and high on the basis of glutenin concentration per g of wheat seed (Table 1). Twenty-two percent (22) of the landraces were placed in the high category for glutenin. The concentration ranged from 17–20 mg/g. wt. The landrace "11757" showed the highest concentration for glutenin (20±0.04 mg/g. wt.). Twenty-seven percent (27) of the landraces were grouped in the medium category with glutenin concentration ranged between 13–16 mg/g. wt. Fifty percent (50) of the landraces with glutenin concentration $\leq 12$ mg/g. wt. were classified in the low category for glutenin. The lowest glutenin content was found in landrace 11618 which was 7.35±1.9 mg/g. wt. (Fig 1, S5 Fig).

### Enzymes activity estimation

**Ascorbate Peroxidase (APX) activity.** The landraces were categorized into three groups (low, medium, and high) based on Ascorbate peroxidase activity per g of wheat seed (Units/g. wt.) (Table 1). Twenty percent (20) of the landraces showed a high APX activity. The activity ranged from 931–1560 Units/g. wt. The landrace "11757" possessed the highest APX activity (1560±113.58 Units/g. wt.). Sixty-two percent (62) of the tested landraces were grouped in the medium category with APX activity ranged between 401–932 Units/g. wt. Seventeen percent (17) of the tested landraces were classified in the low category for APX activity. The lowest APX activity was found in the landrace "11538" (160±31.61 Units/g. wt.) (Fig 1, S6 Fig).

**Catalase (CAT) activity.**    The landraces were categorized into three groups (low, medium, and high) based on catalase activity per g of wheat seed (Units/g. wt.) (Table 1). Forty-five percent (45) of the wheat landraces showed a high catalase activity. The activity ranged from 351–590 Units/g. wt. The landrace "11760" possessed the highest catalase activity (589.5±61.20 Units/g. wt.). Fifty-two percent (52) of the landraces were grouped in the medium category with catalase activity ranged between 101–350 Units/g. wt. Two percent (2) of the landraces with catalase activity ≤100 Units/g. wt. were classified in the low category. The lowest catalase activity was found in the landrace "11562 "which was 72±20.52 Units/g. wt. (Fig 1, S7 Fig).

**Protease (PROT) activity.**    The landraces were categorized into three groups, low, medium, and high based on protease activity per g of wheat seed (Units/g. wt.) (Table 1). Seventeen percent (17) of the wheat landraces showed a high protease activity. The activity ranged from 8,801–20584 Units/g. wt. The landrace "11618" possessed the highest protease activity (20584±942.97 Units/g. wt.). Twenty-four percent (24) of the landraces were grouped in the medium category. The protease activity ranged between 6,001–8,800 Units/g. wt. in this category. Fifty-eight percent (58) of the landraces with protease activity ≤6000 Units/g. wt. were classified in the low category. The lowest protease activity was found in the landrace "11539" (1160±230 Units/g. wt.) (Fig 1, S8 Fig).

**Superoxide Dismutase (SOD) activity.**    The landraces were categorized into three groups, low, medium, and high based on SOD activity per g of wheat seed (Units/g. wt.) (Table 1). Fifty-seven percent (57) of the landraces showed a high SOD activity. The activity ranged from 251–310 Units/g. wt. The landrace "11757" possessed the highest SOD activity (303±16.80 Units/g. wt.). Sixteen percent (16) of the landraces were grouped in the medium category with SOD activity ranged between 171–250 Units/g. wt. Twenty-six percent (26) of the landraces were classified in the low category for SOD activity. The lowest SOD activity was found in the landrace "11560" (83±32.10 Units/g. wt.) (Fig 1, S9 Fig).

**Peroxidase (POD) activity.**    The landraces were categorized into three groups (low, medium, and high) on the basis of peroxidase activity per g of wheat seed (Units/g. wt.) (Table 1). Sixteen percent (16) of the wheat landraces showed a high POD activity. The activity ranged from 25001-32341Units/g. wt. The highest POD activity (32341±1097.82 Units/g. wt.) was observed in the landrace "11624". Seventy-eight percent (78) of the landraces were grouped in the medium category with POD activity ranged between 13001–25000 Units/g. wt. Five percent (5) of the landraces were classified in the low category for POD activity. Lowest peroxidase activity was found in the landrace "11570" which was 8371±201.5 Units/g. wt. (Fig 1, S10 Fig).

**Ascorbic acid (AsA).**    The landraces were categorized in three groups, low, medium, and high based on ascorbic acid content per g of wheat seed (μg /g. wt.) (Table 1). Seventeen percent (17) of the landraces showed a high ascorbic acid content. The content ranged from 682–713 μg /g. wt. The landrace "11757" possessed the highest AsA content (533±36.1 μg /g. wt.). Twelve percent (12) of the landraces were grouped in the medium category with AsA content ranged between 641–681 Units/g. wt. Seventy percent (70) of the landraces were classified in the low category for AsA content. The lowest AsA content was found in the landrace "11607" which was 219±34.6 μg /g. wt. (Fig 1, S11 Fig).

## Principal Component Analysis (PCA)

Data were subjected to principal component analysis. Out of the 10 principal components PC (s), seven viz. PC-1, PC-II, PC-III, and PC-IV had Eigenvalues >1 and contributed for 69.82% of total cumulative variability among different genotypes (S2 Table). The contribution of PC-I towards variability was the highest (33.76%) followed by PC-II (15.48%), PC-III (10.84%), and

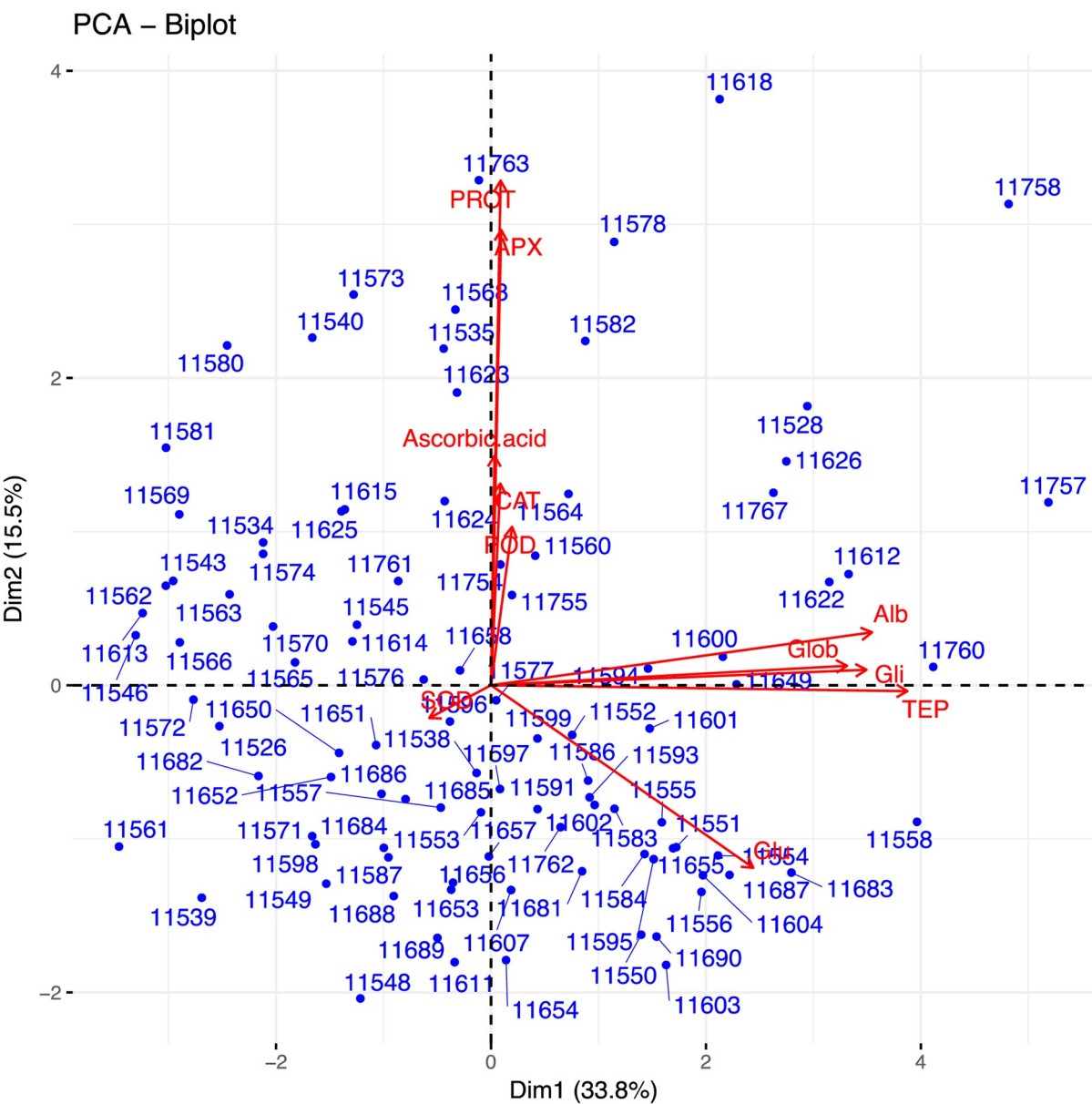

**Fig 2. Bi-plot of wheat genotypes for first two principal components.**

so on. The biplot depicted overall landraces for 11 traits. The first two principal components which contributed 49.25% towards total variance were plotted as biplot on PC-I X axis and PC-II on Y-axis to detect the association between different clusters. The genotype by trait (G-T) biplot thus described 49.25% of the total variation. In G-T biplot, a vector was drawn from origin to every trait which enables the visualization of inter-relationships among characters (Fig 2).

## Discussion

Modern bread wheat cultivars have low genetic diversity compared with wild ancestors because of the domestication of selected cultivars with high yield [21, 22]. Similarly, in Pakistan the

majority of wheat cultivars are built around few cultivars and therefore cultivars are subject to narrow genetic make. To have a sustained wheat production, there is an urgent need to introduce novel diversity.

Wheat genetic resources, especially landraces are the most important means of enhancing diversity and introgression of novel alleles related to end quality parameters into bread wheat [23]. There is a long history of landrace characterization and exploitation to transfer genes of economic traits to current bread wheat varieties. Such, a characterization enables breeders to identify desirable lines for breeding and to devise strategies [24]. Keeping in view these facts, 99 wheat landraces stored in the Plant Genetic Resources Institute (PGRI) were characterized for the end-use quality parameters.

The ability of wheat flour to be processed into different foods is largely determined by the gliadins and glutenin collectively known as gluten proteins [25]. These constitute up to 63–90% of the total grain proteins [26]. Due to their unique viscoelastic properties, gliadin and glutenin are responsible for the bread-making quality of wheat flour [27] as these are present as a continuous cohesive network that surrounds the starch granules [28]. This network is important for bread quality, affecting loaf volume, crumb structure, and initial texture [29]. The albumin and globulin fractions are not known to play a direct role in bread-making; however, they might be necessary for normal baking properties [30]. In comparison with the glutenin and gliadin, albumins and globulins have a better spectrum of essential amino acids (lysine, arginine, aspartic acid, threonine, and tryptophan).

In the present study, the landraces were categorized into three groups based on their seed storage protein concentrations. The three (3) landraces "11558", "11757", and "11760" collected from the Balochistan region showed the highest concentration for all seed storage proteins. While 1 landrace (11593) collected from Gilgit-Baltistan showed medium concentration for all seed storage proteins. Concentrations of all seed storage proteins were lower than already reported for Pakistani cultivars [31, 32]. In our study the concentration of glutenin ranged from 10-20mg/g. wt. and concentration of globulin ranges from 18–21 mg/g. wt. [32] reported high concentrations for glutenin and globulin proteins in Pakistani wheat cultivars. The colossal difference could be due the fact that nowadays farmers are using more and more fertilizer to get maximum yield [33]. It is a well-known fact that foliar spray of N and S fertilizers at anthesis stage influence the expressions of storage proteins genes [34]. However, the high or medium concentrations of seed storage proteins in these landraces are not an attribute of high nitrogen and sulfur fertilizers. All these landraces have been developed in an environment with low nutrients availability. Thus, these landraces represent a source of variation for the development of varieties adapted to cropping systems with low fertilizer input.

Total extracted protein content (TEP) was found higher in 22 wheat landraces. The total protein content determines dough extensibility and elasticity which is very important for chapati-making or baking [35].

Intake of wheat seed proteins in the form of the different products can induce several immune-mediated diseases which include gluten-sensitive enteropathy (celiac disease) [36], Baker's asthma, and wheat-dependent exercise-induced anaphylaxis (WDEIA) in predisposed individuals [37]. The *Triticum aestivum* (wheat) storage protein globulin is a potential food allergen [38], identified as the first candidate wheat protein associated with the development of type 1 diabetes (T1D) [39], and now with celiac disease as well [40]. Similarly, the glutens specifically, gliadins are also known to have epitopes that cause celiac disease [41].

In the current study, we identified a total of 11 wheat landraces showing the low concentration of both glutenin and gliadin. Out of these, 4 landraces were collected from Balochistan while 7 were collected from the Gilgit-Baltistan region. All these landraces with low glutens were also found to be low in total extracted protein. Recently, genetic engineering has been

considered a promising tool to develop low-gliadin wheat varieties, which can be used as raw material for foods for gluten-intolerant people [42, 43]. These wheat landraces with low glutens and total protein contents can be used for the breeding of cultivars with low gliadins and glutenin content.

The ability of seeds and young seedlings to cope with oxidative stress during early vegetative growth, biotic (attachment of the soil and seed-borne pathogens), and abiotic stresses (drought, salinity, heat and chilling) is vital for crop performance and production [44]. High activities of antioxidant enzymes such as superoxide dismutase (SOD), catalase (CAT), peroxidase (POD), and ascorbate peroxidase (APX) have been recorded during seed germination, early growth, biotic, and abiotic stresses [18, 45, 46]. The activities of these antioxidant enzymes are also known to affect the dough quality.

In the current study, a high APX activity was recorded in the seeds of 20 wheat landraces, and high activity of CAT was recorded in 45 wheat landraces. Both APX and CAT catalyze the conversion of $H_2O_2$ into $H_2O$. APX uses ascorbate as an electron donor and CAT uses cofactor iron or manganese [47]. CAT is also involved in oxidative reactions during bread making. The high activities of CAT were recorded in developing kernel for the detoxification of $H_2O_2$ [48]. Both of these enzymes are also known to be involved in wheat salinity and drought stress tolerance. The high activity of APX and CAT was found in Pakistani wheat cultivars Pavon (APX), Pasban (CAT). These cultivars were resistant against drought and salinity stresses [32].

The high activity of PROT was recorded in 17 wheat landraces. PROT plays an essential role in physiology as well as the development of plants. During seed germination, high protease activity mobilizes the stored proteins in seed and acts as a source of amino acids required for the synthesis of novel proteins [49]. Proteases occur naturally in flour. These enzymes break down gluten. This reduces mixing time, making the dough easier to knead, increases dough extensibility, and improves gas retention [50].

The high POD activity was recorded in 8 landraces. Plant peroxidases have an important role in various physiological processes, like POD enhances the process of lignification which is an important defense response against the soil-borne pathogens. It is also involved in cross-linking of pectin, and structural proteins in the cell wall, and catabolism of auxins [51]. The high activity of POD is also linked with improved dough quality by decreasing the adhesiveness of the dough [52]. The high activity of POD was recorded for IQBAL-2000 and BHAKKAR-2000. These wheat cultivars were found resistant for drought and salinity stresses [53, 54].

The high AsA activity was recorded in 14 wheat landraces. AsA interacts directly with superoxide anion radical and hydroxyl radical. AsA acts as a plant growth regulator through hormone signaling. The high activity of AsA in the flour had a marked effect on the gluten network. During the kneading process, the AsA act as an oxidizer giving strength to the gluten network making the dough more elastic [55].

The high SOD activity was recorded in 15 landraces. This enzyme acts as an antioxidant and protects the oxidation of cellular components through reactive oxygen species [56]. The high activity of SOD was recorded in drought-tolerant wheat cultivars Mantar [57].

In the current study, the seeds of wheat landraces showed variation in their ROS scavenging activities. Seeds of landrace "11568" showed high activities for APX, CAT, PROT, and AsA, while medium activity for SOD and POD. Similarly, seeds of wheat landraces "11574" showed high activities for APX, CAT, and PROT, while low activities for SOD and AsA. The landrace "11758" exhibited high antioxidant enzyme activities for APX and CAT, while medium activities for PROT, SOD, and AsA. It has been reported that variations in the activities of these antioxidant enzymes in wheat are genotype specific.

Principal component analysis reveals the chief contributor's significance to the overall variation at each differentiation axis. The Eigenvalues assist in defining the total factors which can be retained. The sum of the Eigenvalues is generally equivalent to the number of variables [57]. Numerals with the highest absolute value nearer to unity in the first principal component affect the grouping more in comparison to those with lesser absolute value nearer to zero [58]. In the present study, out of the 10 principal components PC(s), four viz. PC-1, PC-II, PC-III and PC-IV had Eigenvalues >1 and contributed to 69.82% of total cumulative variability among different wheat landraces. The contribution of PC-I toward variability was highest (33.76%). PC-I showed positive factor loadings for all of the tested parameters except for SOD, while PC-II indicated positive factor loading for all of the tested parameters except for TEP, Glu, and SOD. Usually, one variable/parameter is chosen from these recognized clusters depending on individual loadings. Hence, the highest variation was contributed by TEP which is followed by Albumin in PCI; whereas, PROT followed by APX is a major contribution of variation in PCII. These results clearly show that PC (s) analysis, in parallel to genetic resource characterization pointed out particular traits of interest for designing breeding strategies.

## Conclusion

In Pakistan breeding efforts are focused mainly on increasing the per hectare yield of wheat, thus the potential of grain quality improvement remained unexploited. In the current study, we have explored 99 wheat landraces from the Plant Genetic Resource Institute. We found a great variation in these landraces which can be exploited in breeding programs. The landraces such as "11534", "11539", "11543", "11546", "11561", "11562", "11572", "11580", and "11581" can be used for developing cultivars with the low gluten content. Similarly, the wheat landraces such as "11558", "11757", "11758", and "11760" can be used for developing cultivars with the high seed storage protein.

## Supporting information

**S1 Fig. Means and standard deviation for total extracted proteins concentration in seeds of wheat landraces.**
(TIFF)

**S2 Fig. Means and standard deviation for albumin concentration in seeds of wheat landraces.**
(TIFF)

**S3 Fig. Means and standard deviation for globulin concentration in seeds of wheat landraces.**
(TIFF)

**S4 Fig. Means and standard deviation for gliadin concentration in seeds of wheat landraces.**
(TIFF)

**S5 Fig. Means and standard deviation for glutenin concentration in seeds of wheat landraces.**
(TIFF)

**S6 Fig. Means and standard deviation for APX concentration in seeds of wheat landraces.**
(TIFF)

**S7 Fig. Means and standard deviation for CAT activity in seeds of wheat landraces.**
(TIFF)

**S8 Fig. Means and standard deviation for PROT activity in seeds of wheat landraces.**
(TIFF)

**S9 Fig. Means and standard deviation for SOD activity in seeds of wheat landraces.**
(TIFF)

**S10 Fig. Means and standard deviation for POD activity in seeds of wheat landraces.**
(TIFF)

**S11 Fig. Means and standard deviation for AsA content in seeds of wheat landraces.**
(TIFF)

**S1 Table. Wheat landraces used in this study.**
(DOCX)

**S2 Table. Principal component analysis for selected parameters in wheat landraces.**
(DOCX)

## Acknowledgments

Dr. Aisha Khan, Assistant Professor, Department of Humanities, COMSATS University Islamabad.

## Author Contributions

**Conceptualization:** Sumaira Farrakh.

**Data curation:** Waseem Haider.

**Formal analysis:** Waseem Haider, Maryam Ilyas.

**Funding acquisition:** Sumaira Farrakh.

**Methodology:** Iram Mughal, Yusra Shah.

**Resources:** Muhammad Fayyaz.

**Supervision:** Sumaira Farrakh.

**Validation:** Shamaila Tahir.

**Writing – original draft:** Sumaira Farrakh.

**Writing – review & editing:** Tayyaba Yasmin, Sumaira Farrakh.

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
