## [Decision Letter · Decision Letter 0]

27 Dec 2019

PONE-D-19-25632

Protein Quantification and Enzyme activity Estimation of Pakistani Wheat Landraces

PLOS ONE

Dear Dr farrakh,

Thank you for submitting your manuscript to PLOS ONE. After careful consideration, we feel that it has merit but does not fully meet PLOS ONE’s publication criteria as it currently stands. Therefore, we invite you to submit a revised version of the manuscript that addresses the points raised during the review process.

We would appreciate receiving your revised manuscript by Feb 10 2020 11:59PM. To enhance the reproducibility of your results, we recommend that if applicable you deposit your laboratory protocols in protocols.io, where a protocol can be assigned its own identifier (DOI) such that it can be cited independently in the future. For instructions see: http://journals.plos.org/plosone/s/submission-guidelines#loc-laboratory-protocols

We look forward to receiving your revised manuscript.

Kind regards,

Dorin Gupta, Ph.D.

Academic Editor

PLOS ONE

Journal Requirements:

Additional Editor Comments:

Please address all the suggestions of two reviewers. Revised manuscript should include separate word file of detailed responses to reviewer comments, how and where you made those changes. Reviewers and editors should be able to clearly see changes made in the manuscript as well.

Reviewers' comments:

Reviewer's Responses to Questions

**Comments to the Author**

1. Is the manuscript technically sound, and do the data support the conclusions?

Reviewer #1: Yes

Reviewer #2: Partly

2. Has the statistical analysis been performed appropriately and rigorously? 

Reviewer #1: Yes

Reviewer #2: No

3. Have the authors made all data underlying the findings in their manuscript fully available?

Reviewer #1: Yes

Reviewer #2: Yes

4. Is the manuscript presented in an intelligible fashion and written in standard English?

Reviewer #1: No

Reviewer #2: No

5. Review Comments to the Author

Reviewer #1: Manuscript should be revised to remove the grammatical errors and formatting errors

Abstract-Reads like results summary; need to be concise and brief with highlighting the main findings (line 13-25; page 1)

Introduction

Line 32-35-Please restructure the sentence (page 2)\\

Line 45-48-Need more clarity (page 2-3)

Hypothesis should be there in the introduction section and the whole introduction section should be rewritten to the have a clear understanding and stating the purpose of this study.

Materials and Methods: line 72-pls mention the alterations (page 4)

The entire methodology section seems to be unclear and not enough details provided for its reproducibility; Please rewrite.

Results: PCA section should be explained in detail (page 13)

Discussion: Not framed scientifically; need a thorough revision

Conclusion: Not detailed the main findings/future scope of this work

Reviewer #2: This study quantified the proteins and enzyme activities of 99 wheat landraces of Pakistan. Overall, the protein and enzymes activity variation between landraces can be used in breeding program and helped in decision making for wheat consumption. These findings are quite interesting but their presentation requires many corrections and clarifications. I recommend this publication, but there are major issues that needs to be fixed before it can be accepted for publication.

Abstract

L9-13: Please clearly define the objective and method of study. What was the main purpose of study? Like: 99 wheat landraces collected from four Provinces: A, B, C and D of Pakistan for protein and enzymes characterization.

L13 and 18: There is no need to mention that “The results of”. Protein quantification showed that…..

PCA results should be presented in the abstract. What do you infer based on the PCA? Please mention in the conclusion. Can authors make a statement based on this data that landraces from Baluchistan contained higher protein and enzymes activity than another Provinces?

Authors categorised the landraces in the results section but did not mention in the abstract. I suggest that select one landrace with high protein and enzyme content and, then mention it for breeding program.

Introduction

Please clearly define the differences between proteins and enzymes, and their role.

Please be consistent: L30: 20% and L32: 80 percent.

L35: Reference?

L59-63: Please identify and mention the research gap.

Please mention the factors affecting the wheat grain quality characteristics. I suggest the following sequencing of ideas: importance of wheat – major production areas of Pakistan – quality characteristics of wheat grain – amino acids and wheat grain – factors affecting quality – genetic variability.

Material and Methods

What was the experimental design? Did you perform repeated measures? One sample for multiple parameters (ALB-2 and then ALB-3)? If so then please clearly mention this in the statistical analysis.

How many replicates did you use to measure the ALB etc.?

L68: I suggest to make the Table 1 as a supplementary information. Please mention the storage temperature.

L69: I believe COMSAT University is a huge institute please mention the department.

L72: Did you grind the samples? L73: I suggest to use “ground samples” rather than “powered seeds”.

L102 and 115: please add specification of spectrophotometer (make, model, country).

L149-151: Please include the software specifications. Why didn’t you perform ANOVA? Authors did not mention Tukey HSD significance in any of the result sections. Please mention the scaling criteria for Table 4.

Results and Discussion

In statistical analysis, the authors mentioned that they have included mean ± SD; however, table 2 did not represent any SD values with means. In table 3, Please include scale values (range; Source 22) for low, medium and high protein fractions into the caption.

In enzymes activity estimation: “Landraces were categorised into three groups” I suggest to combine all activities and then find out the overall trend – which landraces show higher percentage of APX, CAT etc. and which landraces show medium and low percentage.

L201: Twenty-seven or 27 please be consistent and use figures.

Rewrite the results of PCA – variability along first axis and second axis and wheat does it mean? and make some meaningful conclusion in the discussion section. 21 landraces showed strong relationship with protein and enzymes activities. Find out the origins of these landraces and discuss those accordingly.

There is no need to repeat the importance of wheat in discussion section. Please report the meaning of your results. For example: Protein quantification suggested that these landraces can be used for breeding program and what are the management practices can increase the quality characteristics of wheat. Adaptation variability of different agro-ecological zones can potentially influence the grain quality and, please mention these kinds of reasoning into the discussion section rather than comparing.

Note: Please focus on the English.

Good luck.

6. PLOS authors have the option to publish the peer review history of their article (what does this mean?). If published, this will include your full peer review and any attached files.

Reviewer #1: No

Reviewer #2: No

---

## [Author Response · Author response to Decision Letter 0]

11 Mar 2020

Subject: Answers to comments of honorable reviewer 1

Honorable reviewer,

Thank you so much for sparing your time for reviewing this article. We have tried our best to remove all short comings of the manuscript according to your suggestion. We have highlighted changes in blue font.

Reviewer #1: Manuscript should be revised to remove the grammatical errors and formatting errors

Ans. Thank you so much, we have tried to remove all grammatical errors

Abstract-Reads like results summary; need to be concise and brief with highlighting the main findings (line 13-25; page 1)

Ans: with due respect, there were 11 parameters that we have studied for these landraces, and mentioned all main find for each parameters.

Introduction

Line 32-35-Please restructure the sentence (page 2)\\

Line 45-48-Need more clarity (page 2-3)

Hypothesis should be there in the introduction section and the whole introduction section should be rewritten to the have a clear understanding and stating the purpose of this study.

Materials and Methods: line 72-pls mention the alterations (page 4)

Ans: Thank you very much, we have changed almost entire Introduction section and mentioned purpose of study.

The entire methodology section seems to be unclear and not enough details provided for its reproducibility; Please rewrite

Ans: we have deleted the confusing statements from the methodology.

Results: PCA section should be explained in detail (page 13)

Ans: we have explained the PCA analysis.

Discussion: Not framed scientifically; need a thorough revision

Ans: we have re-write the discussion

Conclusion: Not detailed the main findings/future scope of this work

Ans: we have re-write the conclusion

Subject: Answers to comments of honorable reviewer 2

Honorable reviewer,

Thank you so much for sparing your time for reviewing this article. We have tried our best to remove all short comings of the manuscript according to your suggestion. We have highlighted changes in Green font.

Reviewer #2: This study quantified the proteins and enzyme activities of 99 wheat landraces of Pakistan. Overall, the protein and enzymes activity variation between landraces can be used in breeding program and helped in decision making for wheat consumption. These findings are quite interesting but their presentation requires many corrections and clarifications. I recommend this publication, but there are major issues that needs to be fixed before it can be accepted for publication.

Abstract

L9-13: Please clearly define the objective and method of study. What was the main purpose of study? Like: 99 wheat landraces collected from four Provinces: A, B, C and D of Pakistan for protein and enzymes characterization.

Ans: correction has been incorporated and highlighted in green.

L13 and 18: There is no need to mention that “The results of”. Protein quantification showed that

Ans: Correction has been incorporated

PCA results should be presented in the abstract. What do you infer based on the PCA? Please mention in the conclusion.

Can authors make a statement based on this data that landraces from Baluchistan contained higher protein and enzymes activity than another Provinces?

Authors categorised the landraces in the results section but did not mention in the abstract. I suggest that select one landrace with high protein and enzyme content and, then mention it for breeding program.

Introduction

Please clearly define the differences between proteins and enzymes, and their role.

Please be consistent: L30: 20% and L32: 80 percent.

L35: Reference?

L59-63: Please identify and mention the research gap.

Please mention the factors affecting the wheat grain quality characteristics. I suggest the following sequencing of ideas: importance of wheat – major production areas of Pakistan – quality characteristics of wheat grain – amino acids and wheat grain – factors affecting quality – genetic variability.

Ans: we have entirely changed the introduction.

Material and Methods

What was the experimental design? Did you perform repeated measures? One sample for multiple parameters (ALB-2 and then ALB-3)? If so then please clearly mention this in the statistical analysis.

How many replicates did you use to measure the ALB etc.?

Ans: The material methods section has been changed. All of the fraction were pooed into one Eppendrof tube before the analysis. for each samples three replicates were used

L68: I suggest to make the Table 1 as a supplementary information. Please mention the storage temperature.

Ans: Table 1 has been mentioned as Supplementary Table.

L69: I believe COMSAT University is a huge institute please mention the department.

Ans: Correction has been done.

L72: Did you grind the samples? L73: I suggest to use “ground samples” rather than “powered seeds”.

Ans:Correction has been done.

L102 and 115: please add specification of spectrophotometer (make, model, country).

Ans: Suggestion has been incorporated

L149-151: Please include the software specifications. Why didn’t you perform ANOVA? Authors did not mention Tukey HSD significance in any of the result sections. Please mention the scaling criteria for Table 4.

Ans: we online program for creating PCA and calculation of mean and SD. We have mentioned the scaling parameters and merge two table to give batter view for the analysis. we did not perform ANOVA, because since the main objective was the estimation of concentration only.

Results and Discussion

In statistical analysis, the authors mentioned that they have included mean ± SD; however, table 2 did not represent any SD values with means. In table 3, Please include scale values (range; Source 22) for low, medium and high protein fractions into the caption.

Ans: Respected reviewer, we have entered the scale under the table and also, mentioned the mean and SD in supplementary figures.

In enzymes activity estimation: “Landraces were categorised into three groups” I suggest to combine all activities and then find out the overall trend – which landraces show higher percentage of APX, CAT etc. and which landraces show medium and low percentage.

Ans: According to respected reviewer suggestion, we have tabulated all findings in one table

L201: Twenty-seven or 27 please be consistent and use figures.

Rewrite the results of PCA – variability along first axis and second axis and wheat does it mean? and make some meaningful conclusion in the discussion section. 21 landraces showed strong relationship with protein and enzymes activities. Find out the origins of these landraces and discuss those accordingly.

There is no need to repeat the importance of wheat in discussion section. Please report the meaning of your results. For example: Protein quantification suggested that these landraces can be used for breeding program and what are the management practices can increase the quality characteristics of wheat. Adaptation variability of different agro-ecological zones can potentially influence the grain quality and, please mention these kinds of reasoning into the discussion section rather than comparing.

Ans: Thank you so much for such valuable suggestions. We have re-written the PCS result and discussion including conclusions.

Note: Please focus on the English.

Good luck.

Thank you so much

---

## [Decision Letter · Decision Letter 1]

19 May 2020

PONE-D-19-25632R1

Protein Quantification and Enzyme activity Estimation of Pakistani Wheat Landraces

PLOS ONE

Dear Dr farrakh,

Thank you for submitting your manuscript to PLOS ONE. After careful consideration, we feel that it has merit but does not fully meet PLOS ONE’s publication criteria as it currently stands. Therefore, we invite you to submit a revised version of the manuscript that addresses the points raised during the review process.

This is an interesting study, however, please address all the comments/suggestions further to bring this manuscript to publication standards. 

We would appreciate receiving your revised manuscript by Jul 03 2020 11:59PM. To enhance the reproducibility of your results, we recommend that if applicable you deposit your laboratory protocols in protocols.io, where a protocol can be assigned its own identifier (DOI) such that it can be cited independently in the future. For instructions see: http://journals.plos.org/plosone/s/submission-guidelines#loc-laboratory-protocols

We look forward to receiving your revised manuscript.

Kind regards,

Dorin Gupta, Ph.D.

Academic Editor

PLOS ONE

Additional Editor Comments (if provided):

Dear Authors thank you for addressing all the comments from reviewers. However, there are further suggestions to be addressed:

There are many grammatical and typographic errors in manuscript, which should be thoroughly corrected. We would recommend that you have your manuscript copy-edited by either a native-English speaking colleague or a professional copy-editing service. While you may approach any qualified individual or any professional scientific editing service of your choice, PLOS has partnered with American Journal Experts (AJE) to provide discounted services to PLOS authors. AJE has extensive experience helping authors meet PLOS guidelines and can provide language editing, translation, manuscript formatting, and figure formatting to ensure your manuscript meets our submission guidelines. If there are still language issues in text that AJE has edited, AJE will re-edit the text for free. To take advantage of this special partnership, visit the AJE website and enter referral code PLOS15 on the registration page for a 15% discount off AJE services (http://www.aje.com/c/plos15). If you are already registered with AJE, please log in and enter PLOS15 at the bottom of your researcher dashboard under ‘Join a Group.’ Please note that PLOS ONE does not receive any compensation in relation to services completed by AJE and that having the manuscript copyedited by AJE or any other editing services does not guarantee selection for peer review.

Authors also need to provide compelling argument to support that existing wheat varieties/advanced breeding lines in Pakistan are low or high in desired different protein types, enzymes and antioxidants. Either provide comparative data from your own studies which suggests you must have used wheat varieties/advanced breeding lines for comparison or support from published literature. For example references which you have cited on page 13- reference 31 doesn't provide needed information, however, reference 32's findings are close to what has been demonstrated in this manuscript. Though, this manuscript has added information on different types of proteins.

Authors should rewrite some of the discussion section by adding more meaningful information to bring the flow of information and related literature. It should not read more like repeat of results. Systematic highlight of how each of the protein/ antioxidant is important from either plant's or human health's perspective. This information is provided

as general overview but not for all sections. The discussion should charily outline how superiority of studied lines is in agreement or disagreement of published research, or how these findings are adding value to existing knowledge as reference 32 is one similar publication.

Page 13, where discussion starts in last paragraph- In the current study....., this whole paragraph is little confusing and requires thorough revision.

Reviewers' comments:

Reviewer's Responses to Questions

**Comments to the Author**

1. If the authors have adequately addressed your comments raised in a previous round of review and you feel that this manuscript is now acceptable for publication, you may indicate that here to bypass the “Comments to the Author” section, enter your conflict of interest statement in the “Confidential to Editor” section, and submit your "Accept" recommendation.

Reviewer #1: All comments have been addressed

Reviewer #2: All comments have been addressed

2. Is the manuscript technically sound, and do the data support the conclusions?

Reviewer #1: Yes

Reviewer #2: Yes

3. Has the statistical analysis been performed appropriately and rigorously? 

Reviewer #1: Yes

Reviewer #2: Yes

4. Have the authors made all data underlying the findings in their manuscript fully available?

Reviewer #1: Yes

Reviewer #2: Yes

5. Is the manuscript presented in an intelligible fashion and written in standard English?

Reviewer #1: Yes

Reviewer #2: Yes

6. Review Comments to the Author

Reviewer #1: I am happy with the corrections and edits made by the authors as per my previous comments. I recommend this paper to be published in this journal.

Reviewer #2: Authors have incorporated all the suggested changes in the manuscript. This is an interesting study and will help in breeding program of the wheat.

7. PLOS authors have the option to publish the peer review history of their article (what does this mean?). If published, this will include your full peer review and any attached files.

Reviewer #1: No

Reviewer #2: No

---

## [Author Response · Author response to Decision Letter 1]

28 Jun 2020

Subject: Answer to reviewer comments.

Respected Editor

Thank you so much for your valuable suggestions. We have tried out best to incorporate your suggestions.

Additional Editor Comments (if provided):

Dear Authors thank you for addressing all the comments from reviewers. However, there are further suggestions to be addressed:

There are many grammatical and typographic errors in manuscript, which should be thoroughly corrected. We would recommend that you have your manuscript copy-edited by either a native-English speaking colleague or a professional copy-editing service. While you may approach any qualified individual or any professional scientific editing service of your choice, PLOS has partnered with American Journal Experts (AJE) to provide discounted services to PLOS authors. AJE has extensive experience helping authors meet PLOSguidelines and can provide language editing, translation, manuscript formatting, and figure formatting to ensure your manuscript meets our submission guidelines. If there are still language issues in text that AJE has edited, AJE will re-edit the text for free. To take advantage of this special partnership, visit the AJE website and enter referral code PLOS15 on the registration page for a 15% discount off AJE services (http://www.aje.com/c/plos15). If you are already registered with AJE, please log in and enter PLOS15 at the bottom of your researcher dashboard under ‘Join a Group.’ Please note that PLOS ONE does not receive any compensation in relation to services completed by AJE and that having the manuscript copyedited by AJE or any other editing services does not guarantee selection for peer review.

Ans: The manuscript is thoroughly read by me and Dr. Waseem Haider PhD from University of Illinois at Urbana–Champaign.

Authors also need to provide compelling argument to support that existing wheat varieties/advanced breeding lines in Pakistan are low or high in desired different protein types, enzymes and antioxidants. Either provide comparative data from your own studies which suggests you must have used wheat varieties/advanced breeding lines for comparison or support from published literature. For example references which you have cited on page 13- reference 31 doesn't provide needed information, however, reference 32's findings are close to what has been demonstrated in this manuscript. Though, this manuscript has added information on different types of proteins.

Authors should rewrite some of the discussion section by adding more meaningful information to bring the flow of information and related literature. It should not read more like repeat of results. Systematic highlight of how each of the protein/ antioxidant is important from either plant's or human health's perspective. This information is provided

as general overview but not for all sections. The discussion should charily outline how superiority of studied lines is in agreement or disagreement of published research, or how these findings are adding value to existing knowledge as reference 32 is one similar publication.

Page 13, where discussion starts in last paragraph- In the current study....., this whole paragraph is little confusing and requires thorough revision.

Ans: Sir, thank you so much, we have thoroughly revised the discussion. Sir, there are very few studies on screening cultivars for quality parameters. Th main focus of breeders is on yield and disease resistance parameters. We designed this study to explore the landraces deposit in gene bank for quality parameters. This study could be helpful for breeders for the selection of landraces.

---

## [Editor Report · Decision Letter 2]

8 Jul 2020

PONE-D-19-25632R2

Protein Quantification and Enzyme activity Estimation of Pakistani Wheat Landraces

PLOS ONE

Dear Dr. farrakh,

Thank you for submitting your manuscript to PLOS ONE. After careful consideration, we feel that it has merit but does not fully meet PLOS ONE’s publication criteria as it currently stands. Therefore, we invite you to submit a revised version of the manuscript that addresses the points raised during the review process.

Please address suggestions for correcting manuscript for typographic and grammatical errors. As some of the corrections are made by authors for correcting typographic and grammatical errors though the manuscript does not meet publication criteria at this stage and further corrections are required. Discussion looks fine now. After accepting all the changes, submit manuscript with new corrections for further review.

We look forward to receiving your revised manuscript.

Kind regards,

Dorin Gupta, Ph.D.

Academic Editor

PLOS ONE

Additional Editor Comments (if provided):

Dear Authors, thank you for addressing most of the comments. However, the manuscript still has many typographic and grammatical errors; references in reference list are not written in consistent style. There are many references which needs correction for consistency of the recommended format. Few corrections were made by authors though not completely. Unfortunately, the paper can not be accepted in its current form, unless suggested improvements are made.

We would recommend that you have your manuscript copy edited by either a native-English speaking colleague or a professional copy-editing service. While you may approach any qualified individual or any professional scientific editing service of your choice, PLOS has partnered with American Journal Experts (AJE) to provide discounted services to PLOS authors. AJE has extensive experience

helping authors meet PLOS guidelines and can provide language editing, translation, manuscript formatting, and figure formatting to ensure your manuscript meets our submission guidelines. If there are still language issues in text that AJE has edited, AJE will re-edit the text for free. To take advantage of this special partnership, visit the AJE website and enter referral code PLOS15 on the registration page for a 15% discount off AJE services (http://www.aje.com/c/plos15). If you are already registered with AJE, please log in and enter PLOS15 at the bottom of your researcher dashboard under ‘Join a Group.’ Please note that PLOS ONE does not receive any compensation in relation to services completed by AJE and that having the manuscript copy edited by AJE or any other editing services does not guarantee selection for peer review.

---

## [Author Response · Author response to Decision Letter 2]

16 Aug 2020

Respected Editor

Thank you so much for your giving us another chance of improving our article. Sir we have tried our best to improve grammatical mistakes as well as English language. Hoping for your kind consideration. Thanks

---

## [Editor Report · Decision Letter 3]

7 Sep 2020

Protein Quantification and Enzyme activity Estimation of Pakistani Wheat Landraces

PONE-D-19-25632R3

Dear Dr. farrakh,

We’re pleased to inform you that your manuscript has been judged scientifically suitable for publication and will be formally accepted for publication once it meets all outstanding technical requirements.

Kind regards,

Dorin Gupta, Ph.D.

Academic Editor

PLOS ONE
---

## [Editor Report · Acceptance letter]

14 Sep 2020

PONE-D-19-25632R3

Protein Quantification and Enzyme Activity Estimation of Pakistani Wheat Landraces

Dear Dr. Farrakh:

I'm pleased to inform you that your manuscript has been deemed suitable for publication in PLOS ONE. Congratulations! Your manuscript is now with our production department.

Kind regards,

on behalf of

Dr. Dorin Gupta 

Academic Editor

PLOS ONE